# SoftManiSim: A Fast Simulation Framework for Multi-Segment Continuum Manipulators Tailored for Robot Learning

**Mohammadreza Kasaei**
School of Informatics
University of Edinburgh, UK
`m.kasaei@ed.ac.uk`

**Hamidreza Kasaei**
Department of Artificial Intelligence, Bernoulli Institute
University of Groningen, The Netherlands
`hamidreza.kasaei@rug.nl`

**Mohsen Khadem**
School of Informatics
University of Edinburgh, UK
`mohsen.khadem@ed.ac.uk`

**Abstract:** This paper introduces SoftManiSim, a novel simulation framework for multi-segment continuum manipulators. Existing continuum robot simulators often rely on simplifying assumptions, such as constant curvature bending or ignoring contact forces, to meet real-time simulation and training demands. To bridge this gap, we propose a robust and rapid mathematical model for continuum robots at the core of SoftManiSim, ensuring precise and adaptable simulations. The framework can integrate with various rigid-body robots, increasing its utility across different robotic platforms. SoftManiSim supports parallel operations for simultaneous simulations of multiple robots and generates synthetic data essential for training deep reinforcement learning models. This capability enhances the development and optimization of control strategies in dynamic environments. Extensive simulations validate the framework's effectiveness, demonstrating its capabilities in handling complex robotic interactions and tasks. We also present real robot validation to showcase the simulator's practical applicability and accuracy in real-world settings. To our knowledge, SoftManiSim is the first open-source real-time simulator capable of modeling continuum robot behavior under dynamic point/distributed loading. It enables rapid deployment in reinforcement learning and machine learning applications. This simulation framework can be downloaded from `https://github.com/MohammadKasaei/SoftManiSim`.

**Keywords:** Simulation Framework, Soft Robotics, Mathematical Modeling.

## 1 Introduction

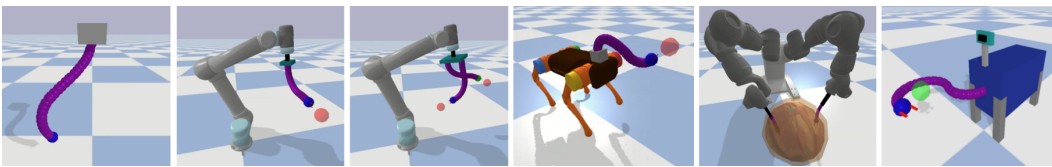

Figure 1: Various simulation scenarios using the SoftManiSim framework and Pybullet demonstrate the versatility of multi-segment continuum robots integrated with different robotic systems. These scenarios highlight the framework's capability to model complex interactions, precise manipulations, and integration with various robotic systems, leveraging advanced mathematical formulations.

Continuum robots, distinguished by their jointless, flexible structures, are pivotal in applications requiring high maneuverability and precision. These robots excel in navigating confined spaces,

8th Conference on Robot Learning (CoRL 2024), Munich, Germany.

Table 1: Overview of existing simulation frameworks and their capabilities.

| Framework | Modeling approach | Physics model complexity | URDF | Rigid robots | Continuum robots | Continuum robot shape information | Adaptive shape resolutions | Support shape extensibility | Rigid-soft hybrid robots | Open source |
|---|---|---|---|---|---|---|---|---|---|---|
| PyBullet/Bullet [17] | rigid-body physics | low | ✓ | ✓ | × | × | × | × | × | ✓ |
| Gazebo [19] | rigid-body physics | low | ✓ | ✓ | × | × | × | × | × | ✓ |
| MuJoCo [20] | rigid-body physics | low | ✓ | ✓ | × | × | × | × | × | × |
| Webots [26] | rigid-body physics | low | × | ✓ | × | × | × | × | × | × |
| Huang et al. [23] | Cosserat rods | medium-high | × | × | ✓ | ✓ | ✓ | × | × | × |
| Elastica [21] | Cosserat rods | medium-high | × | × | ✓ | ✓ | ✓ | × | × | ✓ |
| ChainQueen [27] | Particle-grid hybrid | high | × | × | ✓ | × | × | × | × | ✓ |
| SOFA [22] | FEM | high | × | ✓ | ✓ | ✓ | ✓ | ✓ | ✓ | ✓ |
| PyBullet + SoMo [18] | rigid-body physics | low | ✓ | ✓ | ✓ | × | × | × | ✓ | ✓ |
| **SoftManiSim** | rigid-body physics + Cosserat | low | ✓ | ✓ | ✓ | ✓ | ✓ | ✓ | ✓ | ✓ |

making them suitable for intricate tasks in various settings, particularly in search and rescue operations, exploration, and surgical domains [1]. Several methodologies have been proposed to control such robots, falling into two broad categories: model-based [2, 3, 4, 5] and data-driven approaches [6, 7, 8, 9, 10, 11, 12]. A detailed review of model-based approaches for soft robots is presented in [13]. In recent years, data-driven methods, particularly those leveraging deep reinforcement learning (DRL), have become increasingly popular for controlling soft continuum robots. These techniques excel in managing the robots' complex, nonlinear behaviors and high degrees of freedom, which are challenging to address with traditional control strategies.

DRL approaches rely on large datasets to effectively predict and adapt to a robot's dynamic interactions and deformations in real-time [14, 15, 16]. This capability is crucial for precision-critical applications, such as minimally invasive surgery. Accurate simulation plays a key role in generating the extensive data needed for training DRL models. Thus, the ability to produce detailed and realistic simulation data is essential for leveraging the full potential of DRL. Existing simulators like [17, 18] often represent these robots as a series of discrete, spring-connected links — a method that doesn't fully capture their continuous nature. This simplification leads to significant gaps between how the robots perform in simulations versus real life, reducing their effectiveness in tasks where precision is critical. Recent popular simulation frameworks such as PyBullet [17], Gazebo [19], and MuJoCo [20] are primarily designed for rigid-body dynamics, offering limited support for the nuanced modeling of continuum robots. While more specialized frameworks like Elastica [21], SOFA [22], and Huang et al. [23] address some of these challenges using advanced models like Cosserat rods [24], they often come with increased computational complexity and reduced accessibility.

In this paper, we present SoftManiSim, a novel framework that integrates advanced continuum robot modeling with the robust PyBullet simulator. SoftManiSim combines the simplicity of rigid-body physics with the detailed modeling of Cosserat rods, providing a comprehensive toolset for simulating both continuum and rigid-continuum robots. The Cosserat rod theory is commonly used to model continuum and soft robots [25]. This approach, though accurate, involves solving complex boundary value problems (BVPs), which can be computationally intensive. To address this, we propose a novel solver that rapidly estimates the Cosserat rod equations under dynamic loading without solving the BVP. The solver uses a temporal observer to estimate the robot's initial curvature, transforming the BVP into a parallelizable initial value problem (IVP), thus improving computational efficiency.

The simulator benefits from this solver to offer adaptive shape resolutions, shape extensibility, and leverages PyBullet's powerful physics engine for enhanced simulation capabilities. This integration ensures a balance between accuracy and computational efficiency, making SoftManiSim an ideal platform for developing and optimizing hybrid robotic systems. An overview of existing simulation frameworks and their capabilities is presented in Table 1, illustrating the advantages and versatility of SoftManiSim in addressing the limitations of current methods. Figure 1 demonstrates various simulation scenarios using SoftManiSim framework, highlighting its versatility and capability to model complex interactions and precise manipulations.

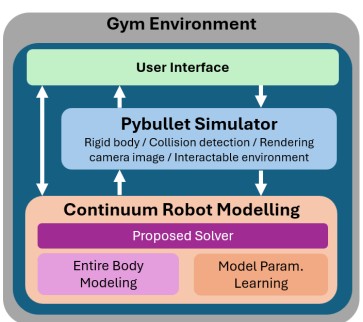

Figure 2: Overview of the framework.

This paper is structured as follows: Section 2 explains the core methodologies employed in the development of SoftManiSim, including the mathematical models

and simulation integration. In Section 3, we detail the approaches for real-time control and policy optimization using our simulation framework. To showcase the efficacy of our simulation and control strategies, a series of simulations are conducted in Section 4, and the results are discussed. Section 5 presents a detailed validation process using real hardware to demonstrate the efficacy of our sim-to-real methodology and the robustness of the control policies developed in simulated environments. Finally, Section 6 concludes the paper with a discussion of the findings, limitations, and potential future work.

## 2   SoftManiSim

This section presents SoftManiSim — a general framework for simulating multi-segment continuum robots. We begin with the mathematical modeling of a single segment which is the core of the proposed framework, extend this to a multi-segment representation, and finally discuss the integration of this model within the PyBullet physics engine to enhance simulation capabilities and interactive functionalities. The overall architecture of this framework is depicted in Figure 2.

### 2.1   Entire Body Shape Modeling

Here, we first use the Cosserat rod equations [24] and follow the approach outlined in [28] to model a continuum robot as a flexible rod. We discuss challenges in solving these equations in real-time and propose a novel solution for rapid estimation of robot shape under dynamic loading. Figure. 3 shows a continuum robot modelled as a rod defined as a curve in space, $r(s) : [0, \ell] \rightarrow \mathbb{R}^3$ and its orientation, $\mathbf{R}(s) : [0, \ell] \rightarrow SO(3)$ as functions of the rod arclength $s \in [0, \ell]$, we can derive the constitutive equations for calculating the instantaneous curvature of the rod $u(s)$ and the overall shape of the rod [29, 28]:

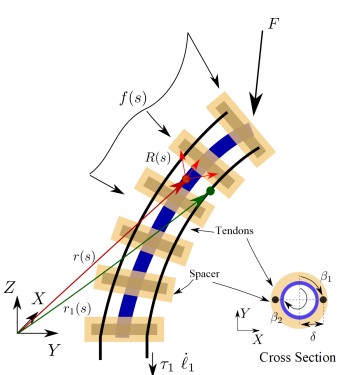

Figure 3: Schematic of a continuum robot with two pull wires under point load $F$ and external distributed load $f$. The robot is composed of two pull wires running in parallel and constrained with respect to each other using spacers. Spacers are fixed to an elastic tube commonly called the main backbone. TDCR curvature can be controlled by pulling the pull wires.

$$r'(s) = \mathbf{R}(s)e_3, \tag{1a}$$

$$\mathbf{R}'(s) = \mathbf{R}(s)[\mathbf{u}(s,t)]_\times, \tag{1b}$$

$$u'(s) = -K^{-1}\Big[[\mathbf{u}(s)]_\times K(u(s) - u^*) + \tag{1c}$$

$$[e_3]_\times \mathbf{R}^T(s)(F(t) + \int_0^s f(t,\sigma)\mathrm{d}\sigma)\Big],$$

$$\ell_i'(s,t) = \|e_3 + [\mathbf{u}(s)]_\times d_i\|, \ i = 1, \cdots, n, \tag{1d}$$

where $'$ denotes a derivative with respect to arc length $s$, the $[.]_\times$ operator is the isomorphism between a vector in $\mathbb{R}^3$ and its skew-symmetric cross product matrix, $e_3 = [0, \ 0, \ 1]^T$ is the unit vector aligned with the z-axis of the global coordinate frame, $u^*$ denotes the pre-curvature of the rod in its reference configuration, $\mathbf{K} = \mathrm{diag}(EI, EI, GJ)$ is the stiffness matrix for the rod, $E$ is the rod's Young's modulus, $I$ is the second moment of inertia, $G$ is the shear modulus, $J$ is the polar moment of inertia, and $F(t)$ and $f(s,t)$ denote the external load at the rod's tip, $n$ is number pull wires.

The model accepts pull wires length as input via boundary conditions. It can be solved using the following boundary conditions:

$$r(0) = [0 \ 0 \ 0]^T, \quad \mathbf{R}(0) = \mathbf{I}_{3\times3}, \tag{2a}$$

$$u_z(0) = 0, \quad \ell_i(0,t) = 0, \quad \ell_i(\ell,t) = L_i(t), \ i = 1, \cdots, n. \tag{2b}$$

where $L_i(t)$ is the desired length of $i^{\text{th}}$ pull wire at time $t$. The model outlined in (1) and (2) form a boundary value problem.

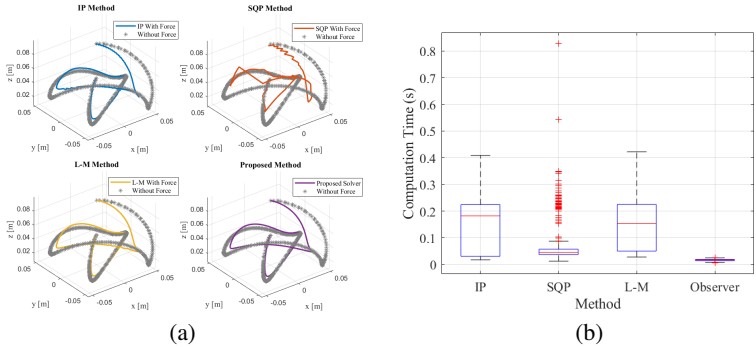

(a)                                                                   (b)

Figure 4: (a) A comparison between the solutions of BVP solvers and the proposed solver for the robot trajectory without external force under the constant curvature assumption. (b) A comparison of computational efficacy between various BVP solvers and the proposed solver.

To solve the equations, it is assumed that at a given time, time-dependent variables (pull wire length and forces) are constant, and the equations are solved in the spatial domain (with respect to $s$) using an ordinary differential equation solver such as Runge–Kutta methods. Next, shooting methods can be used to solve the boundary value problem. A shooting method consists of using a nonlinear root-finding algorithm to iteratively converge on values for $\boldsymbol{u}(0,t)$, in order to satisfy the correct boundary condition for pull wire lengths $L_i(t)$ in (2). Next, the time-dependent variables are updated and the equations are solved again in the spatial domain. Of note, under the assumption of constant curvature bending and no external loading, which is prevalent in the literature [25], curvatures can be analytically estimated from pull-wire lengths to avoid solving the BVP and estimate robot shape in real-time. However, this assumption significantly limits the application of the model.

Our main goal here is to design a solver that will estimate the initial curvature of the robot $\boldsymbol{u}(0,t)$ based on pull wire lengths $L_i(t)$ in real-time to form an IVP system of differential equations and avoid the need to solve the BVP iteratively. The solver modifies boundary conditions into the following initial conditions:

$$\boldsymbol{r}(0) = [0\ 0\ 0]^T, \quad \mathbf{R}(0) = \mathbf{I}_{3\times3}, u_z(0) = 0, \quad \ell_i(0,t) = 0, \tag{3a}$$

$$J(0,t) = \mathbb{I}_{3\times3}, \ D(0,t) = \mathbf{0}_{3\times3\times3}, \ \Gamma(0,t) = \mathbb{I}_{3\times n}, \ \boldsymbol{u}(0,t) = -\int_0^t \mathcal{P}\Gamma^T(\ell,t)\mathcal{V}\epsilon(t)\,\mathrm{d}t. \tag{3b}$$

The new variables introduced in (3b) are

$$\Gamma(s,t) := \frac{\partial \boldsymbol{l}(s,t)}{\partial \boldsymbol{u}(0,t)}, D(s,t) := \frac{\partial R(s)}{\partial \boldsymbol{u}(0,t)}, \quad J(s,t) := \frac{\partial \boldsymbol{u}(s)}{\partial \boldsymbol{u}(0,t)}, \tag{4}$$

and $P(t)$ is the solution of the differential Riccati equation

$$-\dot{P}(t) = -P(t)\Gamma^T(\ell,t)\mathcal{V}\Gamma(\ell,t)P(t) + Q,$$
$$P(t_f) = P_0, \tag{5}$$

$Q$, $\mathcal{V}$, and $P_0$ are all symmetric positive definite matrices. Details of derivation of (3b) are discussed in Appendix A. The model combined with the initial conditions can now be solved as an IVP. To find the shape of the robot, first the equations in (1) are solved with respect to $s$ given the initial values in (3). Then, time dependant variables including $u(0,t)$ and solver's optimal gain $P$ are updated through (3b) and (5), respectively. At the first time step, the initial curvature $u(0,t)$ of the robot is assumed to be zero. For a multi-segment continuum robot, each segment is defined with its own centroids and transformations. By concatenating these transformations and curvature vectors, we model the entire robot's configuration.

Simulations were conducted to evaluate the proposed solver by comparing its predictions with solutions obtained using four different shooting methods: the Interior Point method (IP), the Levenberg-Marquardt method (L-M), and Sequential Quadratic Programming (SQP). These methods, using different root-finding algorithms, are standard BVP solvers. Additionally, we compared the solver's

performance with a model assuming constant curvature, commonly used in the literature. The robot was simulated to follow 500 randomly selected points in the workspace, with root-finding algorithms optimized to a tolerance of $10^{-3}$, and the initial curvature estimate at each sample time used as the starting guess for subsequent steps. Simulation parameters matched those of the actual robot described in Appendix C. In simulations, to model the robot's behavior in the presence of contact, we applied a distributed force of $\mathbf{f} = [1, 1, 1]$ N/m to the robot body and a 3D random force with a magnitude of 0 to 2 N on the robot tip at random angles between $-\pi$ and $\pi$. Comprehensive evaluation of the proposed solver is presented in Appendix B.

Figure 4(a) compares the robot tip position using different solvers with the robot tip position in the absence of force. It is evident that the constant curvature assumption (i.e., neglecting external forces) results in significant errors. Additionally, the proposed solver produces a smoother estimation of the robot tip position compared to other BVP solvers. Figure 4(b) compares the computational efficacy of the solvers. All BVP solvers were set to a tolerance below $1 \times 10^{-4}$ m. As shown, our solver offers the smallest sampling time (15 ms) with the least standard deviation (3 ms). Furthermore, to compare the accuracy of all solvers in terms of satisfying boundary values, we compared the estimated cost. The results showed that our solver provides the best accuracy on par with the L-M method, with the overall boundary value error estimation below $5 \times 10^{-4}$ m. Considering that the best BVP solver, the L-M method, runs at $153 \pm 106$ ms, we can conclude that our solver is approximately 10 times faster than the most accurate solver. More details are provided in Appendix B.

## 2.2 PyBullet Integration

We integrate this model into PyBullet [17], to leverage its potential in creating interactable environments and rendering realistic camera images. This integration is crucial for tasks requiring interaction with environments and visual data processing like visual reinforcement learning, enabling the simulation of visual feedback mechanisms essential for real-world applications like navigation and object manipulation. Additionally, PyBullet's comprehensive suite of tools supports functionalities such as collision detection and real-time environmental adjustments. This fusion not only enhances the accuracy and visual appeal of the simulations but also expands the functional scope of the robot model, making it a powerful platform for detailed monitoring and adaptation in complex scenarios.

## 2.3 Interface and Custom Gym Environment

The `SoftManiSim` class is developed to facilitate the simulation of soft robots using Pybullet physics engine. This class serves as a comprehensive interface that initializes and manages various aspects of the

```
env = SoftManiSim(number_of_segment=5)
shape_world_frame, shape_robot_frame =
    env.move_robot_ori(actions, base_pos
    , base_orin)
```

simulation environment, ensuring a seamless and flexible setup process. To enhance policy learning for continuum robots, we have developed a set of custom Gym environments within our `SoftManiSim` framework, specifically designed to address the dynamic and complex behaviors inherent to these robots. Each environment is defined as a Markov Decision Process (MDP), represented by the tuple $(s_t, a_t, p(s_{t+1}|s_t, a_t), r(s_{t+1}|s_t, a_t))$, where $s_t$ and $a_t$ denote the state and action at time $t$, respectively. The environments are enriched with soft manipulators, rigid obstacles, dynamically moving objects, and other robotic systems, providing a robust and varied context for effective training. Additionally, the gym environments use parallel execution of multiple instances to maximize training efficiency. Detailed information about the interface, its parameters, and more information about the custom Gyms are provided in Appendix D.

## 3 Model-Based Control and Policy Learning

In this section, we explore model-based control techniques and policy learning methods for a multi-segment continuum robot equipped with a dynamically adjustable base.

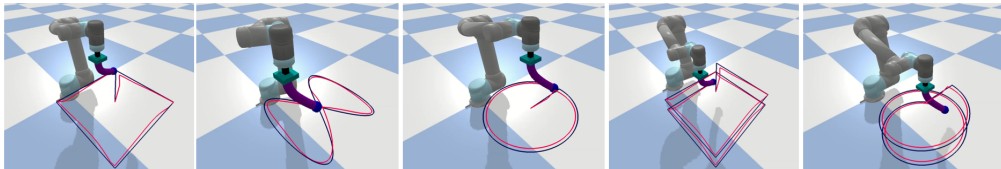

Figure 5: Trajectory tracking results: the robot is tasked with following various paths in both 2D and 3D spaces. The red and black lines indicate the actual and desired paths, respectively.

**Model-Based Control** : Here, we elaborate on the design of a kinematic controller tailored for a multi-segment continuum robot with a dynamically adjustable base. The system's kinematics integrate both segment articulations and base mobility, requiring a comprehensive Jacobian matrix that correlates the robot's end-effector velocity to its configuration space velocities, inclusive of the additional base movements. Utilizing the entire body shape model presented in the previous section, the Jacobian matrix $\mathbf{J}$, which encapsulates the kinematic dependencies, is assumed to accommodate the multi-segment and base mobility features:

$$\dot{x} = \mathbf{J}\dot{q}, \tag{6}$$

where $x = r(\ell, t) \in \mathbb{R}^3$ represents the Cartesian coordinates of the robot's end-effector, and $q \in \mathbb{R}^{n*m}$ represents the control input vector that includes the length of $n$ cables for $m$ segments and the transnational movements of the robot base. The columns of the Jacobian can be estimated using finite difference as

$$J_i = \frac{x^T\left(q + \frac{\triangle q_i}{2}\delta_i\right) - x^T\left(q - \frac{\triangle q_i}{2}\delta_i\right)}{\triangle q_i}, \tag{7}$$

where $\delta_i$ represents each unit vector of the canonical basis in input space, and $\Delta q_i$ denotes small perturbations applied to the $i$-th input to facilitate the numerical Jacobian approximation. The Jacobian matrix supports the development of a robust control strategy, enabling precise manipulation and path-following behaviors.

One can develop a trajectory tracking controller leveraging the pseudo-inverse of the Jacobian matrix, $\mathbf{J}^+$, to guide the robot's end-effector, $x(t)$, along a desired trajectory, $\dot{x}_d(t)$. Utilizing a proportional control law, we aim to minimize the positional error $\tilde{e} = x_d - x$, by adjusting the control inputs based on the error dynamics, $\dot{\mathbf{u}} = \mathbf{J}^+ [\dot{x}_d + \mathbf{K}\tilde{e}]$, where $\mathbf{K}$ is a symmetric positive definite matrix acting as the proportional gain, encompassing both constant and variable components. This control strategy is formulated to effectively zero out the trajectory error, enhancing the robot's accuracy in following the specified path. In the next section, a set of simulations will be conducted to assess the performance of the controller.

**Policy learning:** This method enables the agent to learn from interacting with the environment continuously, aiming to maximize the expected future return $R_t = \mathbb{E}[\sum_{i=t}^{\infty} \gamma^{i-t} r_{i+1}]$ with $\gamma$ as the discount factor. The optimal policy $\pi^*$ is sought, which maximizes the expected return for all states and actions, guided by the Bellman equation. In the next section, we will outline the training scenarios and the policy optimization using customized gym environments based on SoftManiSim, ensuring that the agents can effectively learn and adapt to complex robotic tasks.

## 4   Simulation

Here, a series of simulation scenarios will be designed to evaluate the effectiveness of the proposed framework exploring the diverse strategies for controlling soft and continuum robots.

- **Trajectory Tracking:** The robot follows various trajectories in 2D and 3D spaces, including: **i)** a square on the X-Y plane (0.4 meters sides); **ii)** a figure-eight curve defined by $x = 0.2\sin\left(\frac{2t\pi}{10}\right)$ and $y = 0.2\sin\left(\frac{t\pi}{10}\right)$ for $t$ from 0 to 20 seconds; **iii)** a circle in the X-Y plane (0.2 meters radius); **iv)** a helical trajectory along the Z-axis (0.2 meters radius, 0.1 meters pitch); **v)** a square-helical trajectory along the Z-axis (0.2 meters sides, 0.02 meters pitch).

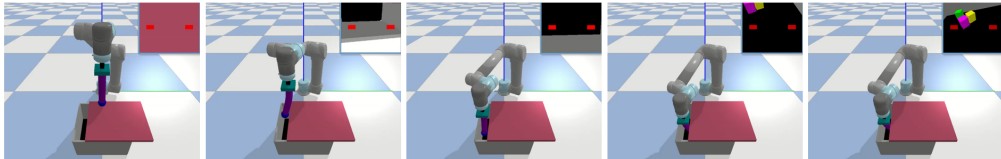

Figure 6: These snapshots illustrate the teleoperation task involving box exploration. In this task, the operator maneuvers the robot through the box and inspects its contents. The operator's camera views are shown in the top right of each image.

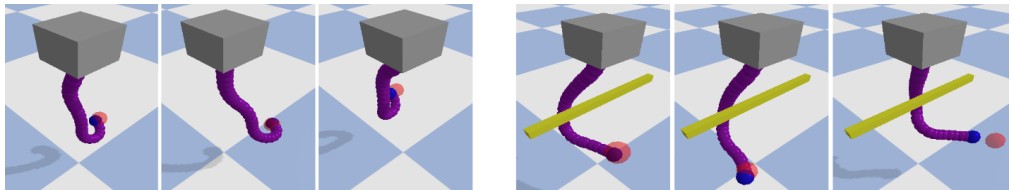

Figure 7: Performance of a Five-Segment Continuum Robot in a 3D Environment. The first three images show the robot reaching a target (red sphere), while the last three depict obstacle avoidance around a yellow bar, highlighting skills developed through reinforcement learning.

- **Box Exploration:** Designed to showcase inspection in confined spaces, a UR5 robot with a two-segment extendable continuum robot (each segment extending up to 0.03 m) is tele-operated via keyboard inputs. The robot navigates through a confined entry into a box, using a camera to inspect the interior. The continuum segments' flexibility and extendability allow thorough inspection, demonstrating precise maneuvers in tight spaces.

- **Reaching Target:** A five-segment continuum robot learns to reach a target in two difficulty levels: **i)** without obstacles; **ii)** with a bar obstacle to avoid. The learning objective is to develop a policy optimizing the path, minimizing distance to the target while avoiding obstacles.

- **Non-Prehensile Object Manipulation:** A three-segment continuum robot, acting as a neck to a quadruped robot (Unitree A1), pushes a cube towards a target. The robot controls its movements using sinusoidal functions, $a \cdot \sin(\omega t)$, where $a$ and $\omega$ are the amplitude and frequency. This setup ensures smooth and precise control, facilitating effective pushing actions.

## 4.1 Results and Discussions

Our evaluation of trajectory tracking for diverse shapes, including square, circle, figure-eight, and helical paths, demonstrates high precision with low mean squared error (MSE). The average MSE for the square, figure-eight, and circle trajectories were $351 \times 10^{-6} m^2$, $409 \times 10^{-6} m^2$, and $195 \times 10^{-6} m^2$, respectively, indicating exceptional accuracy in 2D space. The 3D trajectories, such as the helical and square-helical paths, maintained an MSE of $153 \times 10^{-6} m^2$ and $268 \times 10^{-6} m^2$, respectively, highlighting the robot's capability to execute complex movements. Figure 5 shows representative results (details in Appendix E).

In the box exploration task, initial challenges in maneuvering the continuum robot inside the box were addressed by allowing the operator to disable the movement of rigid body parts, enhancing maneuverability. This adjustment significantly improved the precision and efficiency of internal inspections. In ten tests, the operator successfully completed the task each time, demonstrating the reliability of the control enhancements. Figure 6 illustrates the operational stages and outcomes.

Training a five-segment continuum robot using the Soft Actor-Critic (SAC) algorithm [30] to reach targets with varying difficulty levels showed effective learning and adaptability. The robot successfully reached targets and avoided obstacles, demonstrating proficiency in complex scenarios (Figure 17). Additionally, in non-prehensile object manipulation tasks, the robot effectively pushed a cube towards a target, showcasing its ability to adjust its trajectory and handle the object efficiently (Figure 8). Further results and details are provided in Appendix F.

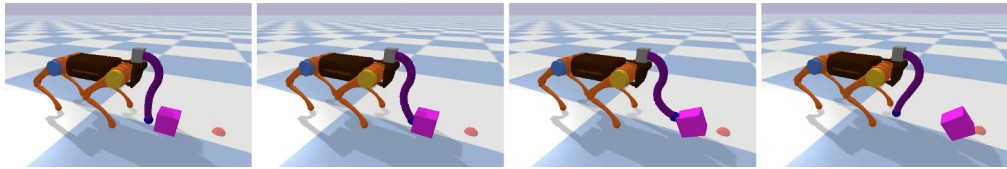

Figure 8: Sequential snapshots showing a quadruped with a three-segment continuum neck, manipulating a cube towards a target (red dot).

The snapshots in Figure 8 highlight a test result where the robot demonstrates its learned skills in non-prehensile object manipulation using SAC. The robot's strategy, driven by a reward function that values both making contact with the object and minimizing the distance to the target, showcases its ability to push the object toward the goal. As the robot progresses through the sequence, it effectively adjusts its trajectory and handling of the cube, leading to a successful alignment with the target in the final snapshot. Further details are provided in Appendix G. A video showing all the results is available online at https://youtu.be/IYqYS4ZQx6k.

## 5  Real Robot Validation

In this section, we explore the potential of SoftManiSim for simulating the behavior of a continuum robot and validating this through real-robot experiments. The scenario involves trajectory-tracking tasks, where the simulated and real robots are programmed to track a set of trajectories. First, we collect data from the real robot, validate the model, and fine-tune the model's parameters, thereby re-

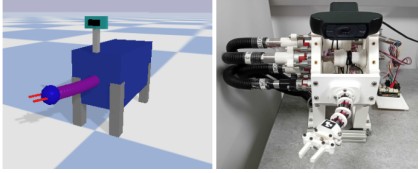

Figure 9: Digital twin and real robot setup.

ducing the sim-to-real gap. Then, we train a control policy in our simulation framework that can be applied to the real robot. Figure 9 shows the flexible soft manipulator used in this experiment and its digital version simulated using SoftManiSim.

After training a control policy and deploying it on the robot, we analyzed the RMSE values summarized in Table 7. These promising results demonstrate the accuracy of our method and validate the effectiveness of the SoftManiSim simulation in closely mirroring real-world dynamics. Detailed information about the robot, experiment setup, experiment details, and more results are provided in Appendix H.

Table 2: Trajectory tracking results

|  | RMSE (mm) | | |
|---|---|---|---|
|  | $\tilde{x}$ | $\tilde{y}$ | $\tilde{z}$ |
| Triangle | 2.87 | 3.14 | 3.08 |
| Square | 3.08 | 3.89 | 3.82 |
| Circle | 1.38 | 1.88 | 2.19 |

## 6  Conclusion and Limitations

In this paper, we introduced SoftManiSim, an innovative simulation framework for multi-segment continuum manipulators. By integrating advanced continuum robot modeling with PyBullet simulator, SoftManiSim offers a comprehensive toolset that bridges the gap between simplified assumptions and the realistic modeling of continuum robots. The framework's ability to support parallel operations and generate synthetic data is crucial for training deep reinforcement learning models, enhancing the development and optimization of control strategies in dynamic environments. Extensive simulations and real robot validations demonstrated the framework's effectiveness, showcasing its capability to handle complex robotic interactions and tasks with high accuracy and computational efficiency.

Despite its advantages, SoftManiSim has several limitations. First, while the proposed solver significantly improves computational efficiency, it may still face challenges in scenarios with extremely high dynamic complexity or when dealing with highly non-linear behaviors not captured by the current model. Additionally, the integration with PyBullet, while robust, might not fully leverage the detailed physics of more specialized soft robot simulation frameworks, potentially limiting the accuracy in some specific cases. Lastly, while our framework supports integration with various rigid-body robots, it may require further optimization and tuning to handle a broader range of hybrid robotic systems seamlessly.

## 7 Acknowledgement

This work is supported by the Medical Research Council [MR/T023252/1].

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

# Appendix A  Real-time and Parallelizable Solver for Continuum Robots

To address the challenges in solving the continuum robot model, we design a nonlinear observer that rapidly estimates the rod's initial curvature $\boldsymbol{u}(s,t)$ without explicitly solving the BVP. This observer transforms the BVP into an initial value problem (IVP) solvable with common ODE solvers. All the parameters and variables used in this section are defined in Table. 3.

To this end, we assume we have an initial estimation for the robot's curvature at $s = 0$ given by $\tilde{\boldsymbol{u}}(0,t)$. Using this value and the first four initial conditions given in (2), we can solve (1). However, as $\tilde{\boldsymbol{u}}(0,t)$ is not accurate, the predicted robot shape and consequently pull wires' length are inaccurate. The error in the estimation of the pull wires' length is

$$\boldsymbol{e}(t) = \tilde{\boldsymbol{l}}(\ell,t) - L_i(t), \tag{8}$$

where $\boldsymbol{l} = [\ell_1, \cdots, \ell_i]^T$ is the vector containing the pull wires' lengths, and $\tilde{\phantom{\cdot}}$ denotes inaccurate estimations due to inaccurate initial curvature.

To design the observer, let us first define an auxiliary variable $\Gamma$:

$$\Gamma(s,t) := \frac{\partial \boldsymbol{l}(s,t)}{\partial \boldsymbol{u}(0,t)}, \tag{9}$$

where $u(0,t)$ is a $3 \times 1$ vector consisting of the initial curvature and the initial twist curvatures of the robot, and $\Gamma$ is a $i \times 3$ matrix, where $i$ is the number of pull wires. Now, we can take the time derivative of the error given in (8) and obtain:

$$\dot{\boldsymbol{e}}(t) = \Gamma(\ell,t)\dot{u}(0,t) - \dot{L}_i(t) \tag{10}$$

| Symbol | Description |
|---|---|
| $\boldsymbol{r}(s)$ | Center line of the robot as a function of arc length $s$ |
| $\mathbf{R}(s)$ | Orientation of the robot as a function of length $s$ |
| $\ell$ | Overall length of the rod |
| $\boldsymbol{u}(s)$ | Instantaneous curvature of the rod |
| $\boldsymbol{e}_3$ | Unit vector equal to $[0, \ 0, \ 1]^T$ |
| $\boldsymbol{u}^*$ | Pre-curvature of the rod in its reference configuration |
| $\mathbf{K}$ | Stiffness matrix equal to $\mathrm{diag}(EI, EI, GJ)$ |
| $E$ | Young's modulus of the rod |
| $I$ | Second moment of inertia |
| $G$ | Shear modulus |
| $J$ | Polar moment of inertia |
| $\boldsymbol{F}(t)$ | External load at the rod's tip |
| $\boldsymbol{f}(s,t)$ | External distributed force |
| $\boldsymbol{d}_i$ | Distance between the pull wire and the rod, see Figure. 3 |
| $\delta$ | Distance from the robot's centroid to the tubes |
| $\beta_i$ | Relative angular position of each pull wire, see Figure. 3 |
| $\ell_i(s,t)$ | Total arc length of each pull wire |
| $L_i(t)$ | Desired length of the $i^{\text{th}}$ pull wire at time $t$ |
| $\tilde{\boldsymbol{u}}(0,t)$ | Initial estimation of the robot's curvature at $s = 0$ |
| $\boldsymbol{e}(t)$ | Error in estimation of pull wires' length |
| $\Gamma(s,t)$ | Auxiliary variable for partial derivatives |
| $\mathcal{V}$ | Symmetric positive definite matrix used in the Riccati equation |
| $P(t)$ | Solution of the differential Riccati equation |
| $Q$ | Symmetric positive definite matrix |
| $[\cdot]_\times$ | Skew-symmetric cross product matrix operator |
| $\epsilon$ | Levi-Civita symbol in three dimensions |
| $D(s,t)$ | Partial derivative of $\mathbf{R}(s)$ with respect to $\boldsymbol{u}(0,t)$ |
| $J(s,t)$ | Partial derivative of $\boldsymbol{u}(s)$ with respect to $\boldsymbol{u}(0,t)$ |
| $\otimes_{n,m}$ | Tensor product operator with contract along $n$ and $m$ |

Table 3: Nomenclature of Variables

Equation (10) is a first-order linear system of equations and can be optimised using the Riccati equations [31] to estimate the initial curvature $u(0, t)$ that minimises the prediction error of the observer $e(t)$ over time. The optimal solution is given as

$$u(0, t) = -\int_0^t P\Gamma^T(\ell, t)\mathcal{V}\epsilon(t)\mathrm{d}t, \tag{11}$$

where $P(t)$ is the solution of the differential Riccati equation

$$\begin{aligned} -\dot{P}(t) &= -P(t)\Gamma^T(\ell, t)\mathcal{V}\Gamma(\ell, t)P(t) + Q, \\ P(t_f) &= P_0, \end{aligned} \tag{12}$$

$Q$, $\mathcal{V}$, and $P_0$ are all symmetric positive definite matrices. At each time step, $e(t)$ and $\Gamma(\ell, t)$ are updated, then $u(0, t)$ is determined by solving (11) and (12).

So far, we have shown that given the value of $\Gamma$, one can design an observer to estimate the initial curvature of the robot and update the robot shape accordingly without the need to solve the robot model iteratively. Now, to find $\Gamma$ we transform the model given in (1) into an observable form. We define some additional partial derivatives, namely, $J$ and $D$ as:

$$D(s, t) := \frac{\partial R(s)}{\partial u(0, t)}, \quad J(s, t) := \frac{\partial u(s)}{\partial u(0, t)}, \tag{13a}$$

We then take a partial derivative of the robot's model in (1) with respect to $u(0, t)$, starting by (1d). We get:

$$\Gamma(s, t) = \frac{(e_3 + [\mathbf{u}(s)]_\times d_i)^T}{\|e_3 + [\mathbf{u}(s)]_\times d_i\|}[d_i]_\times J(s, t) \tag{14}$$

In deriving the above equation, we used the following identities:

$$\frac{d}{dx}(\|g(x)\|) = \frac{g(x)^T}{\|g(x)\|} \cdot \frac{dg(x)}{dx} \tag{15a}$$

$$\frac{\partial([a]_\times b)}{\partial c} = -[b]_\times \frac{\partial a}{\partial c} + [a]_\times \frac{\partial b}{\partial c}. \tag{15b}$$

Equation (14) gives $\Gamma$ as a function $J$, now to estimate $J$ we take partial derivative of (1b ) and (1c) with respect to $u(0, t)$:

$$D' = ([u]_\times^T \otimes_{2,2} D)^T + R \otimes_{2,2} (\epsilon \otimes_{2,1} J), \tag{16a}$$

$$J' = K^{-1}\left[[K(u - u^*)]_\times J - [u]_\times KJ - \right. $$
$$\left. [e_3]_\times D^T \otimes_{2,1} (F(t) + \int_0^s f(t, \sigma)\mathrm{d}\sigma)\right] \tag{16b}$$

where $\epsilon$ is the Levi-Civita symbol in three dimensions defined as

$$\epsilon_{ijk} = \begin{cases} +1 & \text{if } (i, j, k) \text{ is an even permutation of } (1, 2, 3), \\ -1 & \text{if } (i, j, k) \text{ is an odd permutation of } (1, 2, 3), \\ 0 & \text{if any two indices are equal.} \end{cases} \tag{17}$$

In (16), we removed the $(s, t)$ notation for simplicity and in the process of deriving (16), we used the chain rule of differentiation, and the following definitions.

**Definition 1** *Given a vector $x : \mathbb{R}^l$ and a differentiable matrix $M(x) : \mathbb{R}^l \to \mathbb{R}^{m\times n}$, let $N(x) \in \mathbb{R}^{m\times n\times l}$ be a partial derivative of $M(x)$ w.r.t $x$. Then*

$$N_{ijk}(x) = \frac{\partial M_{ij}(x)}{\partial x_k}. \tag{18}$$

**Definition 2** *Let $F(x) : \mathbb{R}^l \to \mathbb{R}^{m \times n}$ and $G(x) : \mathbb{R}^l \to \mathbb{R}^{n \times o}$ be differentiable matrices. Then*

$$\frac{\partial (F(x)G(x))}{\partial x} = \left( G^T(x) \otimes_{2,2} \frac{\partial F(x)}{\partial x} \right)^T$$
$$+ \left( F(x) \otimes_{2,2} \frac{\partial G(x)}{\partial x} \right)^T, \tag{19}$$

*where the transpose operation $^T$ for a 3 dimensional tensor is defined by $F^T_{ijk} = F_{jik}$. Operator $\otimes_{n,m}$ indicates a product of two tensors with $n$ and $m$ denoting which dimensions to contract in each tensor, respectively. For instance, $H = F \otimes_{2,2} G$ is defined as*

$$H_{ijk} = \sum_{l=1}^{n} F_{ilj} \cdot G_{ilk}, \tag{20}$$

Equations (16), (14), and (1) form the generalised model of the robot and can be solved together with the following initial conditions:

$$\boldsymbol{r}(0) = [0\ 0\ 0]^T, \quad \mathbf{R}(0) = \mathbf{I}_{3 \times 3}, \tag{21a}$$
$$u_z(0) = 0, \quad \ell_i(0, t) = 0, \quad \ell_i(\ell, t) = L_i(t), \quad i = 1, \cdots, n. \tag{21b}$$

The initial conditions for the system are:

$$\boldsymbol{r}(0) = [0\ 0\ 0]^T, \quad \mathbf{R}(0) = \mathbf{I}_{3 \times 3}, \tag{22a}$$
$$u_z(0) = 0, \quad \ell_i(0, t) = 0, \tag{22b}$$
$$J(0, t) = \mathbb{I}_{3 \times 3}, \tag{22c}$$
$$D(0, t) = \mathbf{0}_{3 \times 3 \times 3}, \tag{22d}$$
$$\Gamma(0, t) = \mathbb{I}_{3 \times n}, \tag{22e}$$
$$\boldsymbol{u}(0, t) = -\int_0^t \mathcal{P}\Gamma^T(\ell, t)\mathcal{V}\epsilon(t)\,\mathrm{d}t. \tag{22f}$$

The model combined with the initial conditions can now be solved as an IVP.

To find the shape of the robot, first the equations in (16), (14), and (1) are solved with respect to $s$ given the initial values in (22). Then, time dependant variables including $u(0, t)$ and observer gain $P$ are updated through (22f) and (12), respectively. At the first time step, the initial curvature $u(0, t)$ of the robot is assumed to be zero.

## Appendix B Comprehensive Evaluation of the Proposed Solver

Here, we have performed a comprehensive evaluation of the proposed solver within the SoftManiSim framework, benchmarking it against widely used solvers in the field.

As illustrated in Figure 10, traditional techniques solve a variation of the Cosserat Rod model as a boundary value problem (BVP) [32, 33, 34]. This involves a forward pass using ordinary differential equation solvers, such as the Euler Method or Runge–Kutta methods, to estimate the robot's arc parameters. Subsequently, a BVP solver iterates on the solution using an optimization algorithm to find the correct initial condition that satisfies the boundary condition at the robot's end, as indicated by the red arrow. Our novel approach enhances this model by solving the equations as an initial value problem, where the optimal initial curvature is estimated using a bespoke observer. This observer estimates the temporal evolution of boundary values based on the spatial evolution of the robot's arc parameters, as detailed in the paper. This significantly improves the computational efficiency of the solver by eliminating the iterative loop required in BVP solvers. Notably, both methods can solve the problem as an initial value problem in the absence of external forces. However, complexities arise in the presence of contact or forces such as non-constant bending curvature, which are commonly neglected in the literature but essential to bridge the sim-to-real gap in soft robot simulators. This

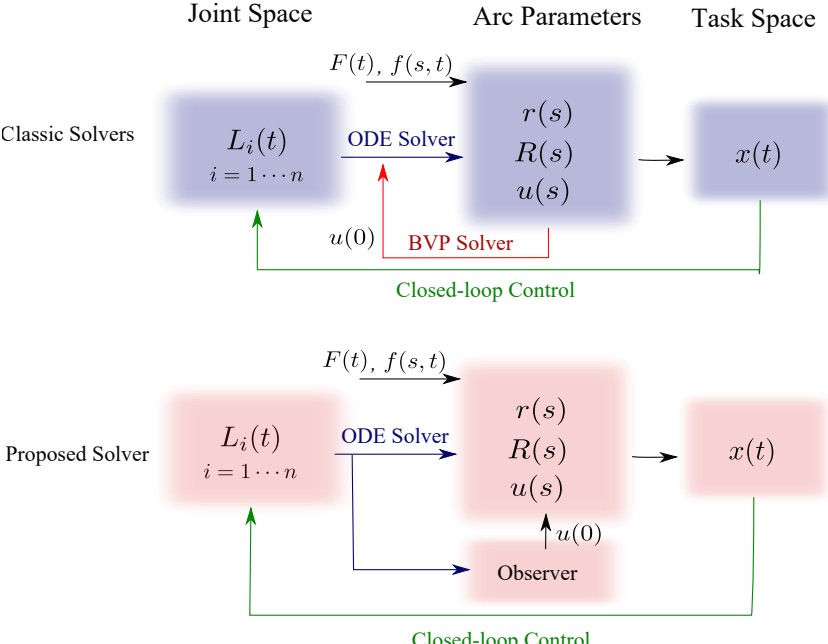

Figure 10: The schematic compares the traditional continuum robot modeling with the proposed solver. Classic techniques either assume the rod bend under constant curvature without any external force/contact or solve a variation of the Cosserat Rod model as a boundary value problem (BVP) to estimate robot tip position defined as $x(t) = r(\ell(t))$ in task space. This involves a forward pass which uses ordinary differential equation solvers such as Euler Method or Runge–Kutta methods to estimate robot arc parameters. Later a BVP solver that uses an optimisation algorithm iterates on the solution to find the right initial condition that satisfies the boundary condition at the robot end. This process is shown with a red arrow. The new approach augments this model to solve the equations as an initial value problem, estimating the optimal initial curvature using a bespoke observer. This significantly improves the computational efficacy of the solver as it includes only a forward pass and avoids the iterative loop used in a BVP solver.

negates one of the most important benefits of soft robots, i.e, capability to interact with environment safely.

To evaluate our method systematically, we first assessed the ODE solver and then the BVP solver. To select the optimal ODE solver, we conducted an experiment by moving a simulated robot (a replica of an exact experimental setup detailed in the paper) across 500 randomly selected points within its workspace. These points and the robot's workspace are shown in Figure 11(a). In the first scenario, to isolate the ODE solver's performance, we assumed no external forces acting on the robot.

We tested state-of-the-art solvers implemented in Python, including the explicit Runge-Kutta (4,5) formula [35], explicit Runge-Kutta (2,3) formula [35], Adams Bashforth Moulton Method [36], and second-order Rosenbrock method [37]. The results show that all non-stiff methods performed similarly, with the Runge-Kutta (2,3) method achieving the fastest sampling time (8.1 ms) and the least standard deviation (1.5 ms). Therefore, we selected it as our main ODE solver for the remainder of the simulations.

Next, we compared our method, which involves only a forward pass, with several existing BVP solvers. All BVP solvers aim to estimate the initial value of the robot curvature, $u(0)$, to minimize the error in the cable lengths, given as an end boundary condition defined by:

$$\text{cost}_{\text{BVP}} = \|\tilde{\mathbf{l}}(\ell, t) - \mathbf{L}(t)\|, \tag{23}$$

where $\|.\|$ denotes the $L_2$ norm, $\mathbf{L}(t) = [L_1, \cdots, L_i]^T$ is the vector of the actual lengths of the pull wires for $i$ wires, $\mathbf{l} = [\ell_1, \cdots, \ell_i]^T$ is the vector of the pull wires' estimated by the model, and $\tilde{}$ indicates inaccuracies due to an incorrect initial curvature.

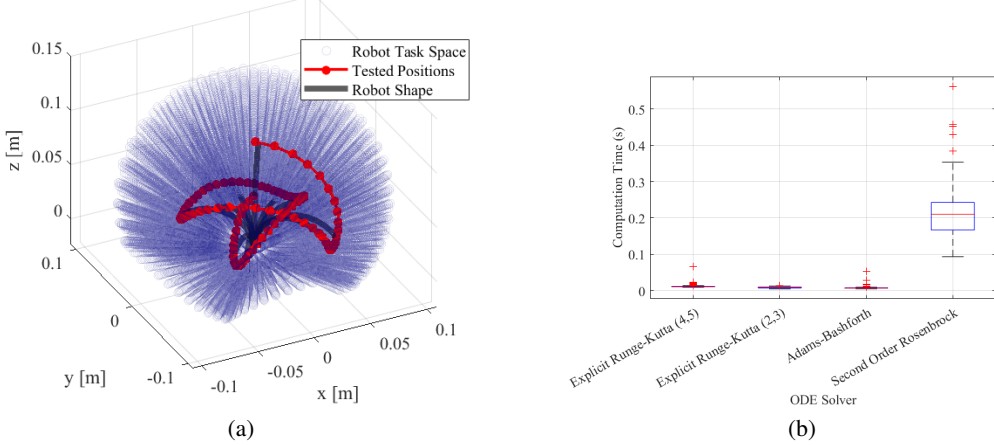

(a)                      (b)

Figure 11: (a) The robot's workspace and 500 randomly selected points. (b) A comparison of the computational efficacy of various ODE solvers across 500 points.

We selected three well-known optimizers to solve the BVP: the Interior Point (IP) method [38], the Levenberg-Marquardt (L-M) method [39], and the Sequential Quadratic Programming (SQP) method [40]. These methods represent the core solvers utilized in many existing papers and simulators [32, 33, 34] and provide a robust baseline for comparative analysis. To ensure a fair comparison, we focused on the performance of the solvers using their standard Python implementations in the SciPy library.

Our benchmarks focused on both computational time and accuracy in estimating the correct boundary value cost function defined in (23). To model the robot's behavior in the presence of contact, we applied a distributed force of $\mathbf{f} = [1, 1, 1]$ N/m to the robot body and a 3D random force with a magnitude of 0 to 2 N on the robot tip at random angles between $-\pi$ and $\pi$. We selected 2 N as the maximum force as forces beyond this value would result in more than 30% strain permanently damaging the robot. The magnitude of the forces for each tested point is shown at the bottom of Figure 12(c). Figure 12(a) compares the robot tip position using different solvers with the robot tip position in the absence of force. It is evident that the constant curvature assumption (i.e., neglecting external forces) results in significant errors. Additionally, the proposed solver produces a smoother estimation of the robot tip position compared to other BVP solvers. Figure 12(b) compares the computational efficacy of the solvers. All BVP solvers were set to a tolerance below $1 \times 10^{-4}$ m. As shown, our solver offers the smallest sampling time (15 ms) with the least standard deviation (3 ms). Furthermore, to compare the accuracy of all solvers in terms of satisfying boundary values, we compared the estimated cost in (23) for each method in Figure 12(c). As can be seen, our solver provides the best accuracy on par with the L-M method, with the overall boundary value error estimation below $5 \times 10^{-4}$ m. Considering that the best BVP solver, the L-M method, runs at $153 \pm 106$ ms, we can conclude that our solver is approximately 10 times faster than the most accurate solver.

These results demonstrate that our proposed solver not only enhances computational efficiency but also improves the accuracy of simulations compared to existing state-of-the-art methods, thereby setting a new benchmark in the field of continuum robot simulation.

## Appendix C    Robot Prototype

Figure 13 depicts the robot used in our experiments, consisting of a flexible backbone stabilized by spacers. At the gripper end, four cables are fixed and pass through the spacers, providing the flexible backbone of the robot. The spacers have enough clearance to follow the curvature of the main backbone. The cables are running in parallel and constrained with respect to each other using

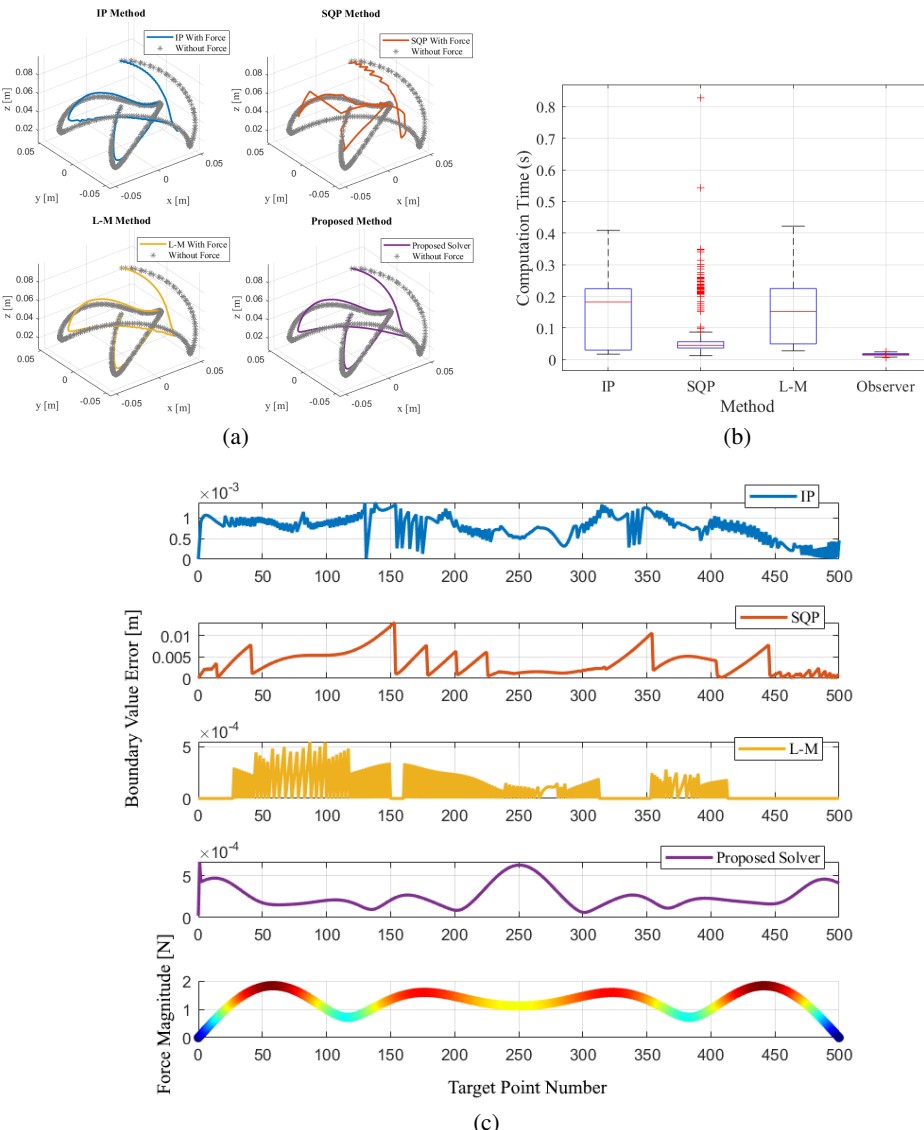

Figure 12: (a) A comparison between the solutions of BVP solvers and the proposed solver for the robot trajectory without external force under the constant curvature assumption. (b) A comparison of computational efficacy between various BVP solvers and the proposed solver. (c) The estimated boundary value error for each method at each of the 5000 tested points in the robot workspace and the magnitude of the force on the robot at each point.

spacers. The backbone curvature can be manipulated by pulling and pushing the cables. The robot is powered by four brushless DC motors from Maxon Motors, each equipped with a quadratic encoder. The motors are controlled by PID position controller modules (EPOS4 Compact 50/5 CAN), which receive feedback from the encoders and interface with a PC via the CAN protocol for setting and retrieving control parameters. A Logitech RGB camera is mounted on the robot's base and for precise location tracking of the robot's tip, an ArUco marker [41, 42] is attached to it, which serves as a critical component in the feedback loop of the control system.

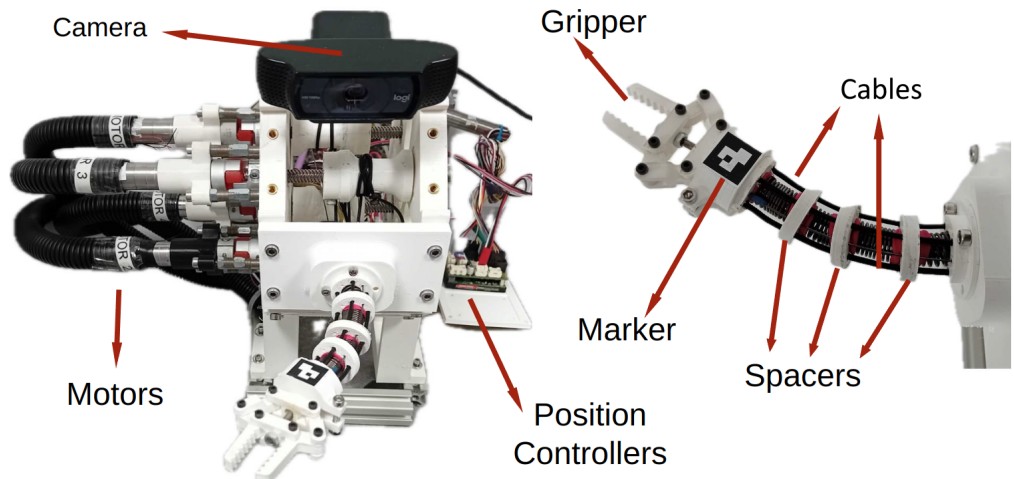

Figure 13: Prototype of our cable-driven continuum robot: The main backbone curvature can be manipulated by pulling and pushing the cables which are controlled by four brushless DC motors, each equipped with a quadratic encoder and position controllers.

Physical parameters of the robot used in the simulations are given in Table 4. In the model verification experiment, the robot's initial length ($\ell(0)$), its second moment of inertia, and polar moment of inertia were measured manually. Later, the robot's tip position calculated from the mathematical model of the robot was compared with the camera measurements at 10 points across the robot's workspace. A least squares algorithm was used to fit model predictions to experimental data to find values of the robot's stiffness and shear modulus. All these parameters are reported in Table 4.

Table 4: Physical parameters of the robot.

| $\ell(0)$ [m] | $I$ [m$^4$] | $J$ [m$^4$] | $E$ [kPa] | $G$ [kPa] |
|---|---|---|---|---|
| 0.07 | 7.363×10$^{-9}$ | 1.4726×10$^{-8}$ | 300 | 70 |

Throughout this work, for all training scenarios, we used the same network architecture and almost the same hyperparameters as described in Table 6.

## Appendix D  Custom Gym Environments

Using SoftManiSim framework, we have developed ten different customized gym environments specifically tailored for continuum robots. Illustrated in Figure 14, these environments are crafted to challenge the robots with a variety of scenarios, each designed to mimic different aspects of real-world applications and the complex physical interactions that continuum robots may face. For detailed examples of how these environments can be applied, please refer to `examples\gyms`.

This development is particularly beneficial for the robotics community as it provides a rich set of tools for testing and refining robotic control policies. By offering a variety of standardized yet challenging scenarios, these environments enable researchers and developers to benchmark and enhance the performance of their robotic systems under controlled but varied conditions. Moreover,

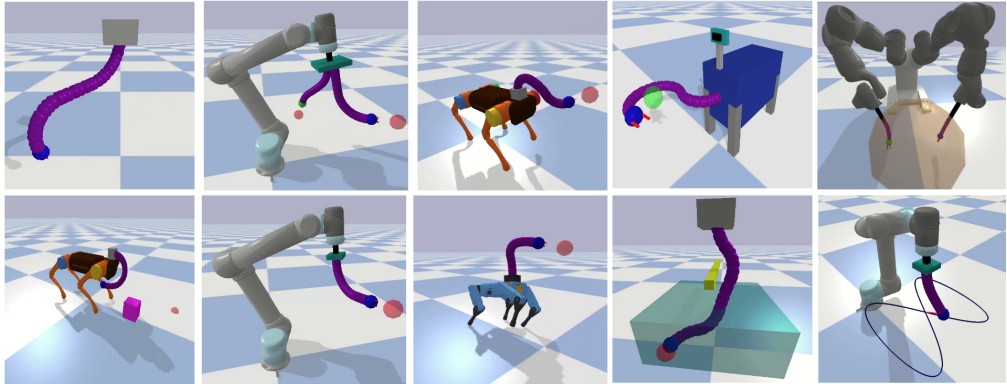

Figure 14: Ten different customized gym environment; To enhance policy learning for continuum robots, we have developed a set of custom Gym environments within our SoftManiSim frameworks that can be used as a baseline.

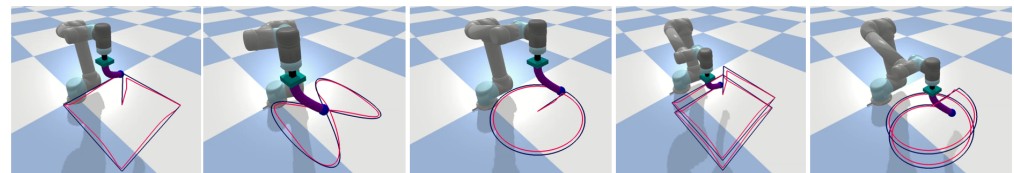

Figure 15: Trajectory tracking results: the robot is tasked with following various paths in both 2D and 3D spaces. The red and black lines indicate the actual and desired paths, respectively.

sharing these resources fosters a collaborative atmosphere within the community, promoting shared learning and accelerating innovation in robotic design and functionality. The availability of these environments ensures that both new and experienced researchers can explore the nuances of robot-environment interaction, thus contributing significantly to the field of continuum robotics.

## Appendix E    Detailed Results of Trajectory Tracking

In this task, a UR5 robot integrated with a two-segment extendable and bendable continuum robot, each segment capable of extending up to 0.03 m, is utilized. The simulated robot was programmed to follow various complex trajectories in both 2D and 3D spaces, designed to test its precision and control capabilities. These trajectories included a square in the X-Y plane with 0.4 meters sides, a figure-eight curve described by specific sinusoidal equations for $x$ and $y$ coordinates over a 20-second period, a circular path with a 0.2-meter radius, a helical trajectory with a 0.2-meter radius and a 0.1-meter pitch, and a square-helical path combining square and helical movements. The effectiveness of the robot's path following was quantitatively assessed by calculating the Mean Squared Errors (MSE) in the X, Y, and Z coordinates for each trajectory. The results, summarized in the provided Table 5 and shown in Figure 15, indicate varied performance across different trajectories. The helical trajectory showed the most precise control, with the lowest average MSE of 0.000153, suggesting that the robot manages consistent vertical movements well. The circular trajectory also exhibited low error rates, emphasizing the robot's ability to maintain steady curvilinear motion. In contrast, the figure-eight and square trajectories had higher MSEs, particularly in the horizontal plane, indicating challenges in managing more complex path changes and corner navigation. The square-helical trajectory achieved a moderate average MSE, highlighting a blend of challenges in maintaining precision in both linear and vertical displacements. These insights can guide further refinements in control algorithms, particularly focusing on improving accuracy in trajectories involving abrupt direction changes and complex geometric patterns.

Table 5: Mean Squared Errors (MSE) for Different Trajectories.

| Trajectory | MSE X | MSE Y | MSE Z | Average MSE |
|---|---|---|---|---|
| Square | 0.000592 | 0.000398 | 0.000062 | 0.000351 |
| Circle | 0.000214 | 0.000341 | 0.000030 | 0.000195 |
| Eight Figure | 0.000930 | 0.000223 | 0.000075 | 0.000409 |
| Helix | 0.000190 | 0.000256 | 0.000014 | 0.000153 |
| Moving Square | 0.000431 | 0.000333 | 0.000041 | 0.000268 |

## Appendix F  Detailed Information About Reaching Target Task

In the reaching scenario, each episode begins with initializing the environment by randomly placing a target within the robot's working area (depicted by green cubes in Figure 17 and Figuire 18), effectively setting a new goal for each session. The length of each episode is set to one, meaning that upon each reset, the robot receives only one observation — the position of the random target — and must immediately decide on an action. This setup compels the robot to rapidly adapt and optimize its policy to minimize the positional error in a single step. The SAC algorithm continuously updates this policy based on the rewards and penalties received, which assess how closely the robot's end-effector reaches the target while remaining within operational bounds. This stringent single-step episode structure accelerates the learning process, demanding high efficiency and accuracy from the robot's decision-making strategies.

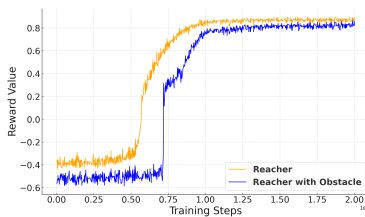

Figure 16: Reward progressions during training.

### Appendix F.1  Reward Function

The reward function for the soft manipulator robot is designed to finely control the robot's behavior in a three-dimensional workspace, across two scenarios: reaching a target and reaching while avoiding an obstacle. The function is defined as:

$$\text{reward} = \text{penalty} + e^{-50 \times (\text{distance}^2)},$$

where distance is the Euclidean distance between the robot's end-effector and the target position. The penalty component is tailored to ensure that the robot operates within its designated bounds and adapts to additional task complexities when an obstacle is present.

For the basic reaching task, the penalty is applied as follows:

$$\text{penalty} = \begin{cases} -0.5 & \text{if } z > 0.28 \text{ or } z < 0.07 \\ 0 & \text{otherwise} \end{cases},$$

this penalty discourages the robot from moving beyond predefined vertical boundaries, effectively reducing the reward when the robot operates outside safe operational zones, thereby enforcing adherence to safe and efficient paths.

In the reaching task with obstacle avoidance, an additional penalty is introduced:

$$\text{obstacle penalty} = \begin{cases} -1.0 & \text{if the robot contacts the obstacle} \\ 0 & \text{otherwise} \end{cases},$$

this ensures that the robot not only aims to reach the target but also learns to navigate around obstacles, further complicating the learning process by penalizing contact with obstacles. Such a mechanism promotes the development of more complex navigation strategies and enhances the robot's ability to handle real-world environments where obstacles are common.

The use of the exponential decay function, $e^{-50 \times (\text{distance}^2)}$, is critical in both scenarios. It creates a strong incentive for the robot to minimize distance to the target, as rewards diminish rapidly with

Table 6: Network architecture and hyperparameters

| Component | Details |
|---|---|
| Actor Network | MLP with 2 hidden layers
Each layer: 256 units, ReLU activation
Output layer: Action size, tanh activation |
| Critic Network | MLP with 2 hidden layers (Twin Critics)
Each layer: 256 units, ReLU activation
Output: Single value (Q-value), no activation |
| Learning Rate | 0.0003 |
| Batch Size | 64 |
| Discount Factor ($\gamma$) | 0.99 |
| Replay Buffer Size | 50000 |
| Number of epoch | $2 \times 10^6$ |

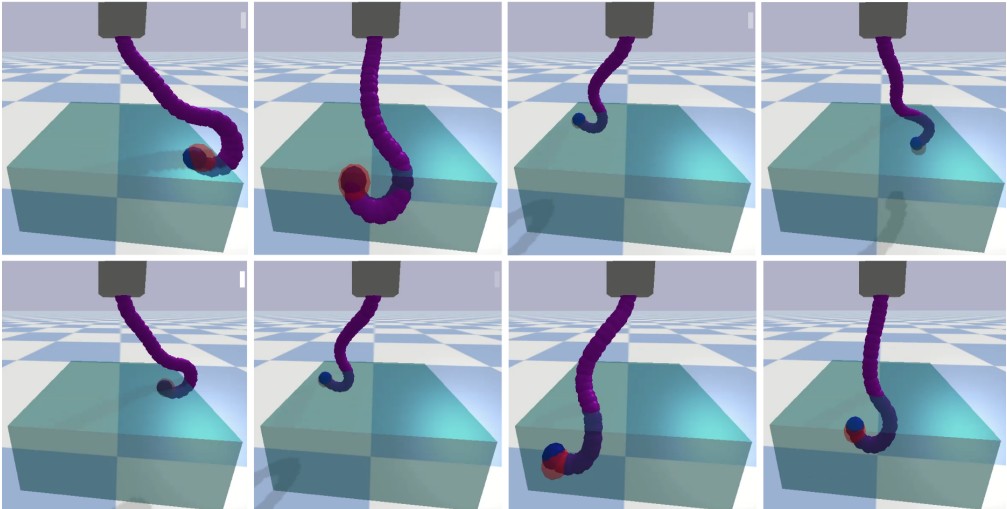

Figure 17: Performance of a five-segment continuum robot in a 3D environment. The robot learned how to reach a target (red sphere), highlighting skills developed through reinforcement learning.

increased distance. This sharp gradient is crucial for reinforcing precise and controlled movements of the end-effector towards the target, thereby playing a pivotal role in the learning algorithm by enhancing the speed and accuracy of the robot's operational capabilities in varied task environments.

Figure 16 shows the progression of mean rewards received during the training of two policies over 2 million steps. This plot illustrates the learning progression for two teacher scenarios: one where the robot solely focuses on reaching a target ("Reacher"), and another where it must reach the target while also avoiding an obstacle ("Reacher with Obstacle"). In both cases, the rewards trend upward, indicating successful learning and adaptation to their respective tasks. However, the introduction of an obstacle in the second scenario introduces a complexity that slightly delays convergence compared to the first scenario. This is reflected in the different trajectories of the reward curves, with the "Reacher with Obstacle" scenario showing a slightly more gradual ascent and later stabilization. The final plateau at a high reward value in both scenarios suggests that the robot effectively learned to reach the target under both sets of conditions, optimizing its path and strategy to maximize the received reward, thereby demonstrating the capability of the reinforcement learning model to adapt to increased task complexity.

In the evaluation of the performance over 100 trials, the mean Euclidean distance between the end-effector and the target was found to be 0.0109 meters with a standard deviation of 0.00471 meters in the scenario without obstacles. This demonstrates a high degree of accuracy and consistency in reaching the target. In contrast, when obstacles were introduced, the average distance increased to

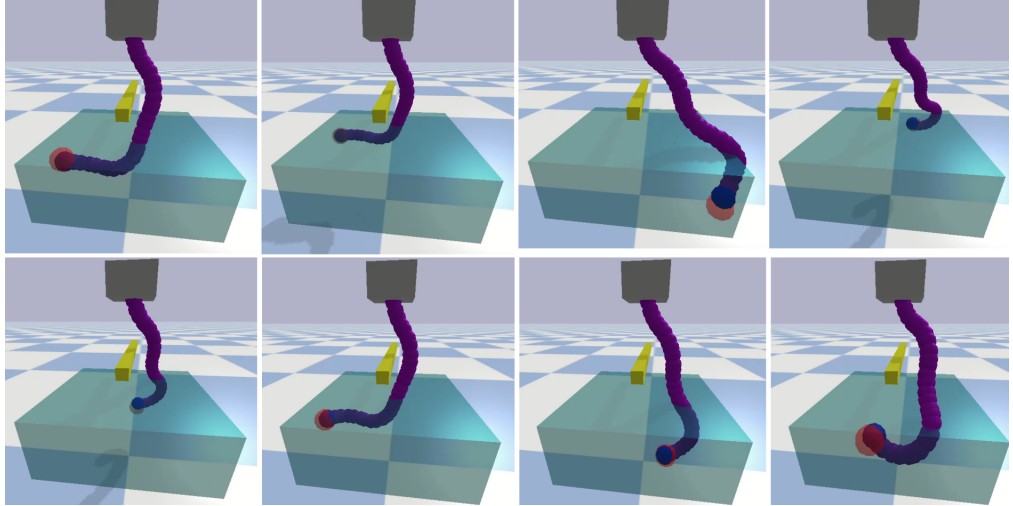

Figure 18: Performance of a five-segment continuum robot in a 3D environment: Demonstrating advanced reaching Skills through reinforcement learning. This robot learned to reach a target (red sphere) while skillfully avoiding a yellow bar obstacle, showcasing its refined abilities acquired from reinforcement learning techniques.

0.0167 meters, and the standard deviation widened to 0.00864 meters. This increase in both the mean and variability indicates a noticeable impact of obstacle presence on the robot's ability to reach the target precisely, reflecting the added complexity and navigational challenges introduced by the obstacles.

## Appendix G    Detailed Results of Non-Prehensile Object Manipulation

In this simulation, a continuum robot integrated onto a Unitree A1 quadruped is tasked with non-prehensile object manipulation, specifically pushing a cube towards a target. During the initialization phase in each run (test/train), the environment is set up which positions the target at a randomly determined location with the x-coordinate between 0.55 to 0.7 meters and the y-coordinate between -0.1 to 0.1 meters, ensuring variability and challenge in starting positions for each trial. The reward function is articulated as follows:

$$\text{reward} = e^{-300 \times (\text{distance\_obj}^2)} + 0.5 \times (\text{touch})$$

where distance_obj is the Euclidean distance to the target, and touch is a binary indicator that adds a bonus if the robot's tip makes contact with the cube, thus encouraging effective interaction with the object.

After training using SAC algorithm, the results demonstrate high precision in the robot's performance. The average absolute errors for 50 trials in reaching the target's x and y coordinates are approximately 0.048 and 0.046, respectively, and the average distance from the target is 0.092 meters. These results highlight the effectiveness of the control strategy in enabling the robot to adapt and accurately manipulate objects towards varying target positions. Figure 19 shows sequential snapshots showing a quadruped with a three-segment continuum neck, manipulating a cube towards a target. A video showing the results is available online at `https://youtu.be/IYqYS4ZQx6k`).

## Appendix H    Real Robot Experiments

### Appendix H.1    Training Dataset

We aim to generate a dataset to validate and to fine-tune our mathematical model. A series of demonstrations was performed by an operator, who adjusted the lengths of various cables to enable

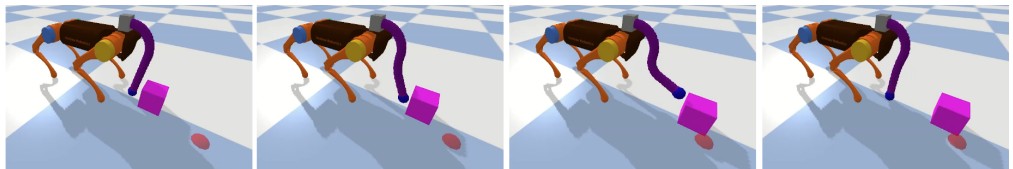

Figure 19: Sequential snapshots showing a quadruped with a three-segment continuum neck, manipulating a cube towards a target (red sphere).

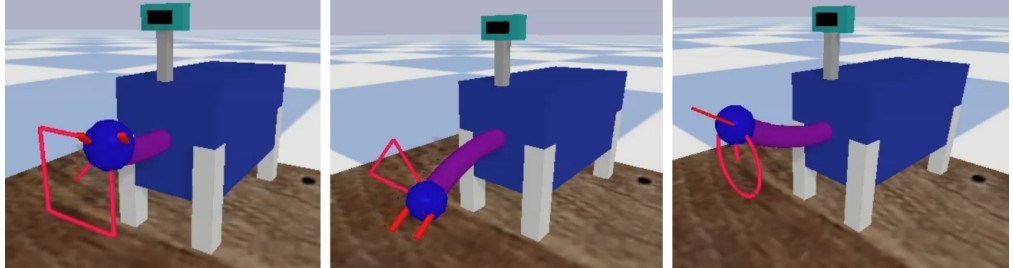

Figure 20: Trajectory tracking results in SoftManiSim: The simulated robot is set to follow a set of predefined trajectories.

the robot's tip to move in multiple directions. Data capturing involved recording both the robot inputs, $\boldsymbol{u_t} \in \mathbb{R}^3$, and the Cartesian coordinates of the robot tip, $\boldsymbol{x_t} \in \mathbb{R}^3$, at a frequency of 15 Hz, forming the training dataset $\mathcal{D} = \{\boldsymbol{x_t^k}, \boldsymbol{u_t^k}\}_{k=1}^{N}$, $N = 100000$. The camera and marker were used to track the robot's position. We employed the collected dataset, comparing the model's outputs with the corresponding targets from the dataset to verify and refine our mathematical model parameters. Subsequently, the mismatches were utilized to train a shallow neural network, designed to address and compensate for any mismatches in the model.

### Appendix H.2    Control Policy Learning

After verifying and refining the mathematical model using the dataset, the next step involves designing a control policy capable of effectively managing the dynamics of the continuum robot. The control policy aims to map observed robot states to actions that drive the robot towards a desired state.

We used our customized gym environment for the robot and SAC algorithm to train a control policy. Each episode begins with a random reset of target positions simulating different starting scenarios and enhancing the robustness of the learning process. The reward function is designed to encourage the agent to minimize the distance between the robot's current end-effector position and the desired position. The reward at each step is calculated as:

$$\text{reward} = e^{-500 \times (\text{distance}^2)}$$

This exponential decay ensures that rewards are higher when the robot's end-effector is closer to the target, providing a strong gradient for learning. After each interaction, the transitions (state, action, reward, next state) are stored in a replay buffer. The SAC algorithm samples batches from this buffer to update the policy and value networks. The learning process involves adjusting the networks to predict more accurate value estimates and to propose actions that maximize these estimates plus the entropy term.

Post-training, the learned policy is validated both in simulated scenarios and real-world tests to ensure its effectiveness. In these simulations and also the real robot experiments, the robot is programmed to follow designated trajectories in two-dimensional space, including: **i)** an equilateral triangle on the XY plane, with each side being 0.04 meters long; **ii)** a square trajectory on the X-Y plane with each side extending 0.025 meters; **iii)** a circular trajectory in the XY plane with a radius of 0.02 meters.

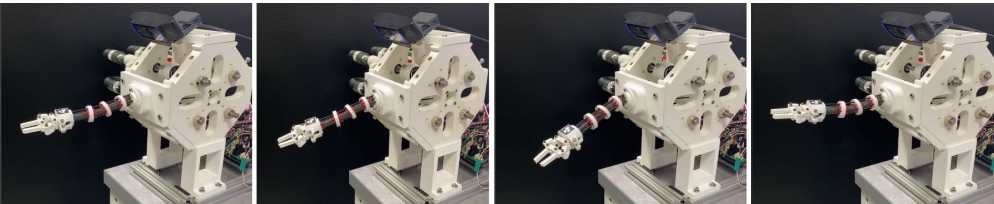

Figure 21: Representative snapshots of the robot while performing the trajectory tracking task (please watch the video at `https://youtu.be/IYqYS4ZQx6k`).

Figure 20 shows the simulation results, as demonstrated, the robot successfully tracked the trajectories. The robot achieved precision with Mean Absolute Errors (MAE) in the x, y, and z directions as follows:

- for the equilateral triangle: 1.77, 1.63, 2.02 mm.
- for the square trajectory: 2.27, 2.03, 2.92 mm.
- for the circular trajectory: 2.54, 1.97, 1.81 mm.

These values demonstrate the robot's accuracy in tracking the designated trajectories, providing a detailed quantitative assessment of the learned policy's effectiveness in both simulated and real-world environments.

### Appendix H.3    Experiments Results

Table 7 presents the Root Mean Square Error (RMSE) measurements in millimeters for trajectory tracking on a real robot, encapsulating its precision across three different geometric paths: triangle, square, and circle. here are the results of trajectory tracking on the real robot:

Table 7: Trajectory tracking results

| | RMSE (mm) | | |
|---|---|---|---|
| | $\tilde{x}$ | $\tilde{y}$ | $\tilde{z}$ |
| Triangle | 2.87 | 3.14 | 3.08 |
| Square | 3.08 | 3.89 | 3.82 |
| Circle | 1.38 | 1.88 | 2.19 |

For the triangular trajectory, the robot exhibited an RMSE of 2.87 mm, 3.14 mm, and 3.08 mm in the $x$, $y$, and $z$ directions respectively, indicating a consistent level of precision across all three axes. The square trajectory showed slightly higher errors, with RMSE values of 3.08 mm in $x$, 3.89 mm in $y$, and 3.82 mm in $z$, reflecting the additional challenges this shape may pose in maintaining accuracy. Notably, the circular trajectory demonstrated the best tracking performance with the lowest RMSE values — 1.38 mm in $x$, 1.88 mm in $y$, and 2.19 mm in $z$ — highlighting the robot's enhanced capability to handle continuous, curvilinear paths with higher precision. Figure 21 shows a set of representative snapshots of the robot while performing this task. A video showing the results is available online at `https://youtu.be/IYqYS4ZQx6k`).

### Appendix I    Python Interface

The `SoftManiSim` class is designed to facilitate the simulation of soft robots using Pybullet physics engine. This class serves as a comprehensive interface that initializes and manages various aspects of the simulation environment, ensuring a seamless and flexible setup process. The constructor of the `SoftManiSim` class takes several parameters, including an optional `bullet` instance, number of segments (`_number_of_segment`), color configurations for the robot's body and head (`body_color`) and (`head_color`), the radius of the body spheres (`body_sphere_radius`), the number of spheres composing the robot's body (`number_of_sphere`), the number of segments in the robot (`number_of_segment`), and a boolean to toggle the graphical user interface (GUI). If no `bullet` instance is provided, the constructor initializes a new Pybullet instance. The `create_robot` method is invoked at the end of the constructor to assemble the robot based on the provided parameters, ensuring that all necessary components are correctly instantiated and configured. This

methodical and thorough initialization process makes the `SoftManiSim` class a powerful tool for researchers and developers, offering a high degree of control and customization over the soft robot simulation, ultimately contributing to more efficient and accurate experimental setups in the field of soft robotics.

### Appendix I.1 API Documentation

Below is the API documentation for the `SoftManiSim` class, detailing essential methods, their arguments and functionalities:

Table 8: API Descriptions of SoftManiSim

| Method | Argument | Description |
|---|---|---|
| `__init__` | bullet | Optional physics engine instance, defaults to None, initializes PyBullet if not provided. |
| | body_color | RGBA color for the robot's body. |
| | head_color | RGBA color for the robot's head. |
| | body_sphere_radius | Radius of spheres used to build the robot's body. |
| | number_of_sphere | Number of spheres constructing the robot's body. |
| | number_of_segment | Number of segments in the robot's body. |
| | gui | Boolean to toggle graphical interface, defaults to True. |
| `create_robot` | - | No arguments, sets up the robot's physical structure within the simulation. This function is invoked at the end of the constructor. |
| `move_robot_ori` | action | Array of actions defining movement commands for robot segments. |
| | base_pos | The base position of the robot in the simulation space. |
| | base_orin | The base orientation of the robot, specified as Euler angles. |
| | camera_marker | Boolean to display camera markers, defaults to True. |
| `calc_tip_pos` | action | Array of actions affecting the tip's position and orientation. |
| | base_pos | The base position from which the tip's calculations start. |
| | base_orin | Base orientation affecting the tip's calculation. |
| `capture_image` | removeBackground | Boolean to decide whether to remove background from the image, defaults to False. |
| `in_hand_camera_capture_image` | - | No arguments, captures image from the robot's in-hand camera. |
| `is_robot_in_contact` | obj_id | Object ID to check for contact with the robot. |
| `is_gripper_in_contact` | obj_id | Object ID to check for contact with the robot's gripper. |
| `suction_grasp` | enable | Boolean to enable or disable the suction grasp mechanism. |
| `set_grasp_width` | grasp_width_percent | Percentage of maximum grasp width to set for the gripper. |
| `add_a_cube` | pos | Position to place the cube in the simulation. |
| | ori | Orientation of the cube, given as a quaternion. |
| | size | Dimensions of the cube. |
| | mass | Mass of the cube. |
| | color | RGBA color of the cube. |
| | textureUniqueId | Optional texture ID for the cube's surface. |
| `wait` | sec | Duration in seconds to delay the simulation. |

## Appendix J Comparison with SOFA

SOFA is a general-purpose solver that uses Finite Element Method (FEM), which fundamentally differs from the direct solution of continuum mechanics equations provided by our fast model. We performed simulations to highlight these differences. We implemented our robot in SOFA in two different versions: a tendon-driven robot with multiple pull/push wires and a simple rod without pull/push wires. The second model is an exact replica of the Cosserat rod plugin for FEM-based Cosserat rod simulation in SOFA [43].

For both models, all physical parameters, including Young's modulus, shear modulus, and physical dimensions, were set equal to the model identified in the paper. The stiffness matrix for both models was $K = \mathrm{diag}[EI, EI, GJ] = \mathrm{diag}[0.0022, 0.0022, 0.001]$. In the simulation, we applied a point load varying from 0 to 0.7 N along the $y$-axis of the rod in its global frame. A load of 0.7 N was found to be the maximum allowable force, as forces higher than this caused a bend of more than 60°

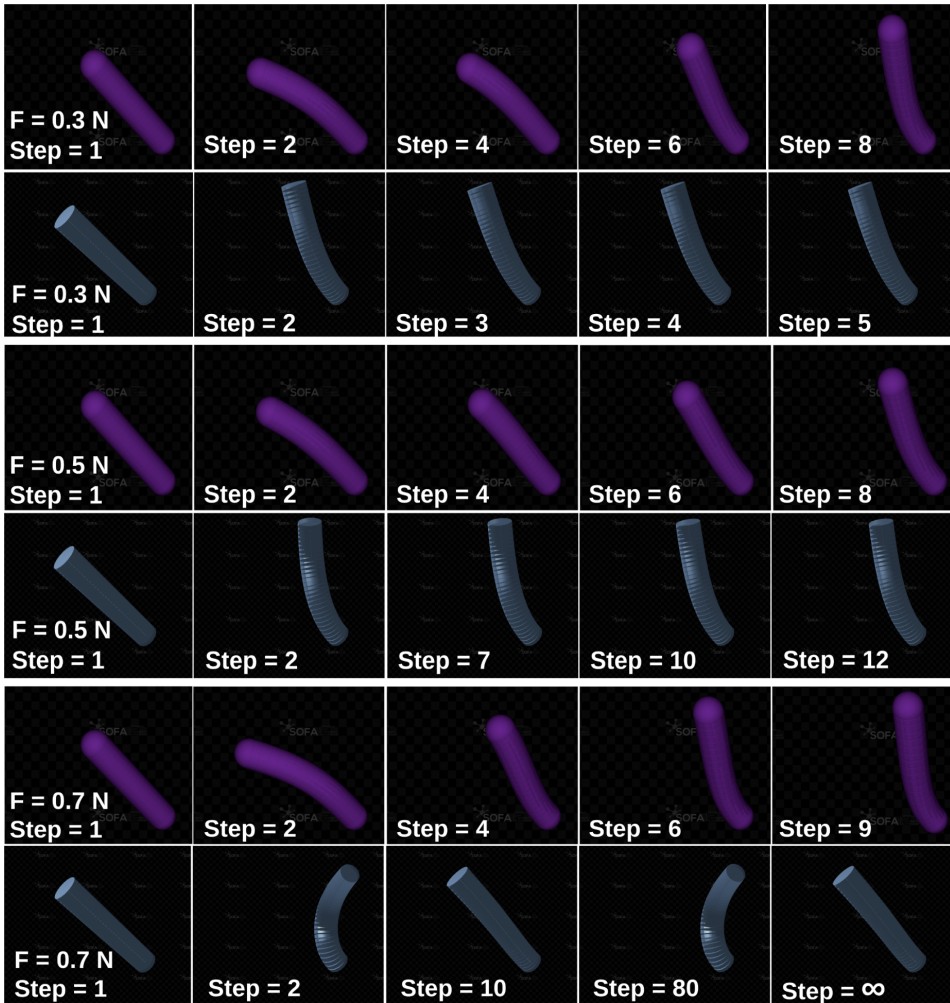

Figure 22: A comparison between the proposed solver embedded in SOFA shown in purple and SOFA's native Cosserat rod plugin.

around the $x$-axis at the rod's tip. This bend causes the projection of the force on the rod's neutral axis to exceed the critical buckling load, leading to instability in the rod.

The critical buckling load $P_{cr}$ for a rod with one end fixed and the other end free can be approximated by Euler's formula:

$$P_{cr} = \frac{\pi^2 EI}{(KL)^2}, \tag{24}$$

where $E$ is the Young's modulus of the material, $I$ is the second moment of area (area moment of inertia), $L$ is the length of the rod, and $K$ is the effective length factor. For a fixed-free end condition, $K = 2$.

Given the defined rod parameters, the critical buckling load for the rod with a fixed-free end condition is approximately $0.542$ N. This can occur when the tip bending angle approaches 50 degrees with a force of 0.7 N along the global $y$-axis (leads to $0.7\text{N} \times \sin(50) = 0.5$ N along robot neutral axis). It should be noted that Euler's formula is an approximation; however, our results also confirm that the maximum allowable force before singularity is 0.7 N.

Results of the simulation are shown in Figure 22. As can be seen, our solver embedded in SOFA is capable of solving the equations up to the singularity with computational consistency. However, SOFA's results are highly dependent on selected meshes and time steps. We were able to perform

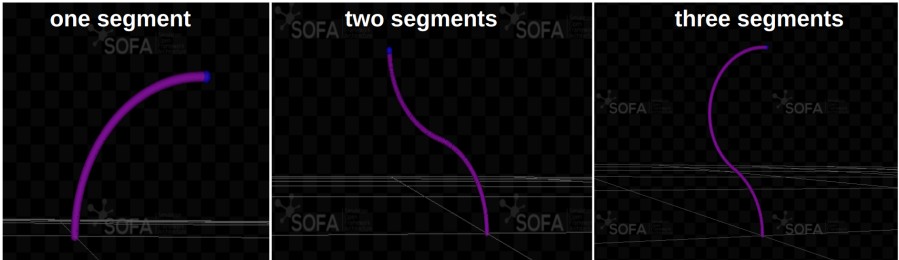

Figure 23: Multi-segment continuum robots simulated using SOFA based on our proposed solver.

simulations on SOFA's Cosserat plugin up to 0.6 N, beyond which it becomes unstable as evident in Figure 22. Additionally, the results' convergence is highly dependent on the number of meshes and sampling times. Table 9 summarises the simulation results, comparing number of samples and overall time required to solve the equations for both methods. Our solver consistently solves the equations at maximum of 9 time steps for all force values including forces close to singularity. SOFA's computational efficiency is variable and depends on meshes and force magnitudes. Finally, the simulation revealed another minor issue with general FEM-based libraries like SOFA. In our model, we have direct access to all the robot's arc parameters (e.g., curvature and pose at any arclength), allowing seamless use of these parameters during modeling to define constraints or apply forces. For instance, we can easily apply a force that is always tangent to the robot tip by measuring its orientation. However, implementing such forces in SOFA proved to be quite complex, requiring the use of constraints and specific solvers, which we struggled to get to converge.

This simulation highlights the differences between our solver and SOFA:

1. **Computational Cost:** FEM can be computationally expensive, especially for large-scale problems, high-resolution meshes, or highly nonlinear problems such as continuum robots under time-varying forces. This makes real-time applications difficult and resource-intensive, which is highlighted in their manual as well [44]. The computational performance of SOFA, which is an FEM-based solver, is not comparable with ours.

2. **Complex Setup and Implementation:** FEM requires careful meshing, which can be complex and time-consuming. Poor mesh quality can lead to inaccurate results or convergence issues.

3. **Numerical Stability and Convergence:** FEM can suffer from convergence issues, especially in highly nonlinear problems involving large bending or complex boundary conditions (e.g., under forces), requiring careful tuning of numerical methods and parameters.

4. **Generalization:** FEM is highly versatile and general, but this can be a drawback for methods like ours that directly solve Cosserat rod equations. However, to provide these benefits, we have implemented our solver in SOFA and made the code available online, so it can be used in more general settings, specifically in contact with other soft objects modeled in SOFA.

Table 9: A comparison between proposed solver embedded in SOFA and SOFA's Cosserat rod plugin.

| Force [N] | SOFA | | SoftManiSim | |
|---|---|---|---|---|
| | Avg. step | Avg. time [s] | Avg. step | Avg. time [s] |
| 0.20 | 5 | 0.05 | 8 | 0.08 |
| 0.40 | 7 | 0.07 | 8 | 0.08 |
| 0.50 | 12 | 0.12 | 8 | 0.08 |
| 0.60 | 24 | 0.24 | 9 | 0.08 |
| 0.70 | × | × | 9 | 0.09 |

As discussed, the proposed solver for the robot is imported into the SOFA simulator. Creating a robot under dynamic loads with SOFA is very cumbersome and requires defining various rods in tandem with specific constraints. However, ours is seamless as the solver can handle any type of rods in any configuration. Figure 23 shows a set of snapshots of SOFA simulating multi-segment continuum robots based on our solver.

