# OpenReview forum: "SoftManiSim: A Fast Simulation Framework for Multi-Segment Continuum Manipulators Tailored for Robot Learning"
_robot-learning.org/CoRL/2024/Conference — CoRL 2024_

### Official Review · Reviewer_osAG · 2024-07-20
**A novel simulator framework with questionable relevance**

**Originality:** 2
**Technical Quality:** 2
**Clarity Of Presentation:** 3
**Potential Impact:** 3
**Recommendation:** 3
**Confidence:** 4

**Review:**

In general, the paper is well written, and the motivation, approaches, and results are mostly clearly communicated. The issues of having accurate and fast simulators for soft robots is a timely issue which requires more research to enable further advances in modeling and control of soft robots. Therefore, the work addresses a significant challenge and is relevant for the community.

Unfortunately, the paper does not make a convincing case for the proposed framework. Starting with showing the limitations of existing frameworks, I expected that a thorough analysis of the accuracy and computational time will be presented.  However, only a case study is presented which did not convince me that it wouldn’t work with one of the other existing framework. In fact, in the introduction it is stated that some existing frameworks  “often come with increased computational complexity and reduced accessibility”. However, there is no numerical evaluation or theoretical analysis of this claim. Instead, the authors implemented some control scenario where I don’t see the relevance to the main contribution of introducing a new simulator. Furthermore, section 5 lacks details on the training data set, the validation, etc.

I suggest focusing on the performance of the simulator by carrying out a thorough analysis of the computational time and accuracy in comparison to the listed existing frameworks. Without that, the relevance of the presented simulator can hardly by evaluated.

UPDATE AFTER REBUTTAL: Increased score

**Quality Of The Limitations Section:**

3

**Questions For Rebuttal:**

Please address the issues mentioned in the “review” section

**Robotics Focus:**

4

**Summary Of Paper:**

In this paper, the authors introduce a novel real-time simulation framework for continuum soft robots, called SoftManiSim. Real time simulation of soft robots is always challenging as their flexible structure results in time-consuming simulations, leading to a trade-off between accuracy and computational time. The authors claim the existing frameworks have some issues and, thus, introduced this novel framework using cosserat rods and Pybullet to overcome these limitations.

**Summary Of Recommendation:**

The paper introduced a novel simulation framework but lacks any detailed analysis or numerical evaluation so that the relevance is questionable.

---

### Official Review · Reviewer_Entu · 2024-07-21

**Originality:** 3
**Technical Quality:** 4
**Clarity Of Presentation:** 4
**Potential Impact:** 3
**Recommendation:** 3
**Confidence:** 5

**Review:**

The paper introduces a novel framework for introducing continuum robot dynamics into conventional robot physics engines (PyBullet, in this case). The paper is generally clearly written and the introduction of Coserat rod continuum robot modeling to this simulation environment is quite interesting as it has not been done before. Depending on the usability of the released package, I think that it will have a good impact on the learning for the soft robots community.

## Strengths
- The work connects the continuum robot community to the robot simulation community, which currently seems to be largely disjointed.
- The authors outline the method well with clear definitions for the modeling methods.
- The authors show diverse simulation environments with different rigid robots in the scene.

## Weaknesses

- The validation on a real-world robot is a bit lacking. The authors present a result on a trajectory tracking task with no perturbation and on a single robot. It's also hard to tell how sensitive the simulator is to simulation parameters. So to me, it is a bit difficult to see how much work it would take to introduce a soft robot into the framework.

- The practical limitations of the work are not very well outlined beside a paragraph saying that it has limitations on scenes with "high dynamic complexity." It would help to understand what these conditions would be where the framework may fail and where the community should work toward filling the gap.

- Many different kinds of continuum robots have been presented in the literature; however, the authors present a specific kind (constant cross-section continuum robot with spacers) for modeling and validation. For example, in (Liu, et al., "Influence of Antagonistic Tensions on Distributed Friction Forces of Multisegment Tendon-Driven Continuum Manipulators With Irregular Geometry"), the authors present one that has a non-constant cross-section with significant tendon friction. It is not clear whether this framework can apply to these robots as well. If it cannot be generalized to these other soft robots, I think that the framework's impacts would be very limited.

- The continuum robotic modeling as outlined in Section 2 is pretty well-established in the community. The contribution here is largely in the integration of the modeling to the PyBullet environment. And I'm not sure if the authors do a convincing enough job comparing against other simulation environments that allow for tendon/continuum robots on performance metrics besides whether or not they have the same specific features that this framework has.

**Quality Of The Limitations Section:**

1

**Questions For Rebuttal:**

- The authors claim that "SoftManiSim is the first open-source real-time simulator capable of modeling continuum robot behavior under dynamic point/distributed loading." However, I believe SofaGym is a pretty well-established tool for this. Could the authors define their contributions more clearly?

- Could the authors shed some light on the clear limitations of this simulation framework's application? What kind of soft robots can this be applied to?

- The authors show sim2real results for a relatively simple trajectory tracking example. I do not think this is the most exciting result to see because I have seen similarly good results from simulators/models relying on piece-wise constant curvature assumptions. Could the authors show some results with more complex environments?

- The authors should present some comparison results on comparing accuracy and runtime against other simulation environments (e.g.,SOFA) that do continuum robot simulations with contact modeling. I get that it is indeed novel that the SoftManiSim allows for rigid robots to be also in the environment but in practice, the rigid robots can usually be abstracted away by directly controlling the base 6DoF pose of the soft robotic manipulator. So more concrete performance comparisons on the soft robot part of this would be helpful.

- How do the speed (dynamic effects) and external contact affect the simulator's accuracy w.r.t the real world?

- How sensitive is the framework to simulation parameters, as well as parameters related to material properties?

**Robotics Focus:**

4

**Summary Of Paper:**

The authors present an open-source simulation framework that allows for both rigid and soft robotic/continuum robotic manipulators. The authors also validate the real-world use case for the framework with a trajectory tracking example on a tendon-driven continuum robot.

**Summary Of Recommendation:**

The paper is generally clearly written and the introduction of Coserat rod continuum robot modeling to this simulation environment is quite interesting as it has not been done before. I have some concerns about the applicability and the usability of the framework in the contexts beyond what was presented in the paper as the paper results are just limited to just one type of a continuum robot.

---

### Official Review · Reviewer_Vw2X · 2024-07-28
**SoftManiSim**

**Originality:** 2
**Technical Quality:** 3
**Clarity Of Presentation:** 3
**Potential Impact:** 2
**Recommendation:** 3
**Confidence:** 3

**Review:**

The paper provides a new implementation of the commonly used cosserat rod model of continuum manipulators. The pybullet implementation allows single and multi-segment manipulators to be combined with rigid robot bases, and provides a basis for policy learning in a relatively accurate environment.

The demonstrate the capacity of the method to be used for policy learning in positioning and manipulation tasks in simulation, and give a limited demonstration of its sim2real transfer. The simulation tasks qualitatively  demonstrate a breadth of capabilities of the method, however no rigorous bench-marking is provided, hence its hard to say whether it exceeds state of the art performance.

The sim2real section is notably quite sparse on details, making it very challenging to assess the quality of validity of the experiment. This section feels very much like it has been inserted as a checkbox exercise to satisfy the need for robot demonstration

Technically the work seems solid, however its contributions are relatively incremental, the formulation is primarily an implementation of known equations in a new environment. Whilst the authors have demonstrated task learning in this environment, it does not substantial extend what is achievable using existing methods

**Quality Of The Limitations Section:**

2

**Questions For Rebuttal:**

1) Could you clarify the major contributions of this paper? I note the authors make some fairly lofty claims about the
2) How does this method compare to other state of this art methods quantitatively? Could you benchmark on time/accuracy?
3) In table 1, benefits of this methods are listed as including "Continuum robot shape information", "Adaptive shape resolutions" and "Support shape extensibility", Could you elaborate on what these mean?
4) In section 5, very little information is given about how the tests were conducted, could you clarify the procedure? i.e how was the model validation and fine tuning performed, what was the speed of movement etc

**Robotics Focus:**

4

**Summary Of Paper:**

SoftManiSim provides a pybullet implementation of cosserat rod equations, enabling fast, accurate learning of multi-segment continuum/hybrid manipulators

**Summary Of Recommendation:**

The work is technically strong and of use to the field as a piece of engineering, but does not demonstrate substantial novelty

---

### Author Rebuttal · Authors · 2024-08-13

We greatly appreciate your insightful feedback and the time you have invested in reviewing our work. We are committed to thoroughly addressing each of the points you have highlighted, and we are confident that these revisions will significantly enhance the quality of our research. We are hopeful that our efforts will be reflected in an improved evaluation.

---

### Decision · Program_Chairs · 2024-09-04

**Decision:**

Accept

**Comment:**

This paper received mixed reviews, leaning to negative.

Strengths:
Paper is generally well written, and well structured
Problem is well motivated
Faster soft simulation is a good research aim with significant interest in the field


For Weaknesses, the reviewers largely focused on a lack of benchmarks, and a lack of comparison to other simulators.  Providing a thorough analysis of the computational time and accuracy in comparison to the listed existing frameworks would significantly strengthen the paper.  As it stands, we currently do not know how well the approach compares to state of the art, and furthermore cannot say that the approach is worthwhile or useful to the community.

The authors should present some comparison results on comparing accuracy and runtime against other simulation environments (e.g.,SOFA) that do continuum robot simulations with contact modeling , including concrete performance metrics.

More detail is required to show how the tests themselves were conducted - e.g., model validation, tuning.

Some sim2real experiments would also be welcomed - e.g.
--How do the speed (dynamic effects) and external contact affect the simulator's accuracy w.r.t the real world?
--How sensitive is the framework to simulation parameters, as well as parameters related to material properties?



Update

The paper has been imrproved and the authors have been responsive to reviewer comments.